# Association of measured quality with financial health among U.S. hospitals

**Samuel J. Enumah**[1]*, **Andrew S. Resnick**[1,2], **David C. Chang**[3,4]

**1** Department of Surgery, Brigham and Women's Hospital, Boston, Massachusetts, United States of America, **2** Department of Quality and Safety, Brigham and Women's Hospital, Boston, Massachusetts, United States of America, **3** Department of Surgery, Massachusetts General Hospital, Boston, Massachusetts, United States of America, **4** Codman Center for Clinical Effectiveness in Surgery, Boston, Massachusetts, United States of America

* samuel.j.enumah@gmail.com

**Data Availability Statement:** Data cannot be shared publicly because of the agreement with the American Hospital Association regarding access to their data. The Hospital Compare data are publicly available from Medicare & Medicaid Services

## Abstract

### Background

High-quality care is a clear objective for hospital leaders, but hospitals must balance investing in quality with financial stability. Poor hospital financial health can precipitate closure, limiting patients' access to care. Whether hospital quality is associated with financial health remains poorly understood. The objective of this study was to compare financial performance at high-quality and low-quality hospitals.

### Methods

We performed a retrospective observational cohort study of U.S. hospitals using the American Hospital Association and Hospital Compare datasets for years 2013 to 2018. We used multilevel mixed-effects linear and logistic regression models with fixed year effects and random intercepts for hospitals to identify associations between hospitals' measured quality outcomes—30-day hospital-wide readmission rate and the patient safety indicator-90 (PSI-90)—and their financial margins and risk of financial distress in the same year and the subsequent year. Our sample included 20,919 observations from 4,331 unique hospitals.

### Results

In 2018, the median 30-day readmission rate was 15.2 (interquartile range [IQR] 14.8–15.6), the median PSI-90 score was 0.96 (IQR 0.89–1.07), the median operating margin was -1.8 (IQR -9.7–5.9), and 750 (22.7%) hospitals experienced financial distress. Hospitals in the best quintile of readmission rates experienced higher operating margins (+0.95%, 95% CI [0.51–1.39], p < .001) and lower odds of distress (odds ratio [OR] 0.56, 95% CI [0.45–0.70], p < .001) in the same year as compared to hospitals in the worst quintile. Hospitals in the best quintile of PSI-90 had higher operating margins (+0.62%, 95% CI [0.17–1.08], p = .007) and lower odds of financial distress (OR 0.70, 95% CI [0.55–0.89], p = .003) as compared to hospitals in the worst quintile. The results were qualitatively similar for the same-year and lag-year analyses.

(https://www.cms.gov/Medicare/Quality-Initiatives-Patient-Assessment-Instruments/HospitalQualityInits/HospitalCompare).

**Funding:** SE was supported by the National Institutes of Health Research Training in Alimentary Tract Surgery T32 Fellowship Award (T32DK007754). https://www.massgeneral.org/surgery/gastrointestinal-and-oncologic-surgery/education-and-training/research-training-in-alimentary-tract-surgery-t32-fellowship The funders had no role in study design, data collection and analysis, decision to publish, or preparation of the manuscript.

**Competing interests:** The authors have declared that no competing interests exist.

## Conclusion

Hospitals that deliver high-quality outcomes may experience superior financial performance compared to hospitals with poor-quality outcomes.

## Introduction

Quality improvement provides tangible benefits to patients [1, 2]. However, quality improvement activities require substantial financial investments on the part of hospital leadership. For example, the National Surgical Quality Improvement Program requires hiring at least one full-time nurse abstractor and paying an annual participation fee [3]. Hospitals vary in how much money they spend on infection preventionists, safety specialists, medical directorships, and additional roles and resources designed to improve quality. Across inpatient quality improvement activities, hospitals may spend between $2 million to $21 million ($200 to $400 per discharge or 1% to 2% of total operating revenue) each year [4]. However, these expenses may negatively affect hospitals' profit margins and create additional economic risks, and hospital leadership may not be certain that these investments in quality make financial sense in a competitive marketplace [5].

Despite the clinical advantages of high-quality care, there remains uncertainty regarding the financial performance of hospitals that deliver excellent quality care. One possibility is that high-quality hospitals may enjoy increased revenue by acquiring additional patients through reputation or by avoiding negative revenue implications as would be imposed by the Hospital Readmissions Reduction Program (HRRP) or other quality-based payment programs (e.g. Hospital Value-based Purchasing Program, Hospital Acquired Conditions Program) [6–9]. Outside of revenue adjustments, additional financial incentives to improve quality may arise from reduced operating expenses, as quality improvement can help avoid process inefficiencies, overuse, and preventable harms [10]. An alternative possibility is that these investments do not offer revenue gains or lower expenses. It is not clear that the Hospital Compare website provides sufficient information for patients to make condition-specific decisions nor is it evident that the revenue adjustments of the value-based quality programs are substantial enough to drive managerial change (e.g. the HRRP penalty is limited to a maximum of 3% for a hospital's quality performance across selected conditions) [7, 11]. Regarding operational expense reduction, the implementation of process efficiencies or adoption of appropriate use of care may be more costly than anticipated or expected financial results may not materialize. Thus, hospital managers may not demonstrate the financial viability of quality strategies within a clearly identifiable timeframe, and they must continuously defend their efforts to invest in improving the quality of patient care.

Given the debate around investments in quality and financial performance, the aim of our study was to explore the association between measured quality and hospital financial health. We sought to explore two aspects of financial health: margins (profit divided by revenue) and financial distress—a comprehensive metric of an organization's financial well-being. First, an improved understanding of the association between measured quality today and an organization's present and future financial margins may provide leaders with valuable evidence to support investments in quality that deliver value to patients. Second, avoiding financial distress— a state in which an organization cannot meet its financial obligations [12]—helps hospitals avoid bankruptcy and closure, an outcome that leaves some patients without access to timely care [13]. If an association exists between measured quality and financial distress, hospital management may see quality as a potential avenue to keep their doors open. We hypothesized

that higher quality was associated with better financial margins and lower odds of financial distress.

## Materials and methods

### Data and study population

We extracted general and financial information from the American Hospital Association (AHA) Annual Survey for the years 2013 to 2018. The AHA conducts an annual national survey of U.S. hospitals and provides data on a broad range of topics including utilization, revenue, structure, personnel, managed care contracts, and information technology [14]. For measured hospital quality, we extracted risk-adjusted readmission rates and patient safety performance scores from Hospital Compare, a publicly available dataset published by the Centers for Medicare & Medicaid Services (CMS) that provides consumers with information on hospital quality at more than 4,000 Medicare-certified hospitals [15]. We linked the AHA Annual Survey database to the Hospital Compare dataset using the Medicare Provider Identification Number. Our population included general short-term adult acute care hospitals that reported at least one year of data. Not all hospitals report quality data to Hospital Compare because certain hospitals do not meet the minimum denominator threshold (twenty-five cases during the reporting period). Details of missingness for the main dependent and independent variables are included in **S1 Table in** S1 File. We included public (non-federal government) and private (for-profit and non-profit) hospitals. We excluded psychiatric and pediatric hospitals and long-term care facilities as these facilities have different reimbursement structures compared to acute care hospitals and may not have similarly defined quality goals. Additional details about the exclusion criteria are included in the (see **S1 Fig in** S1 File). This was a retrospective observational cohort study of hospitals with a repeated measures design, and we followed the Logical Explanations and Visualizations of Estimations in Linear Mixed Models (LEVEL) reporting guideline [16].

### Variables

The main dependent variables were operating margin, total margin, and financial distress. We defined operating margin as operating income divided by net patient revenue. Total margin includes non-operating income and was defined as net income divided by total revenue. The data were trimmed at the $2.5^{th}$ and $97.5^{th}$ percentiles to reduce the influence of outliers. To define financial distress, we used the modified Altman Z-score, a metric that captures a more comprehensive picture of an organization's financial position and incorporates liquidity, profitability, efficiency, and leverage [17]. The modified Altman Z-score is a validated tool to predict the likelihood of hospital bankruptcy [18]. To generate a Z-score, we used the following formula: $Z = 6.56*X1 + 3.26*X2 + 6.72*X3 + 1.05*X4$, where X1 = working capital / total assets, X2 = retained earnings / total assets, X3 = earnings before interest and taxes (EBIT) / total assets, and X4 = total equity / total liabilities. We defined financial distress based on the annual modified Altman Z-score, and, consistent with previous literature, hospitals with a Z-score less than 1.80 were considered financially distressed [18, 19].

The two independent variables of interest were the 30-day hospital-wide risk-standardized readmission rate and Patient Safety Indicator-90 (PSI-90) score. We defined quality based on reported hospital outcomes, and we used the hospital-wide readmission rate and the PSI-90 score—metrics that are risk-adjusted by CMS using patient characteristics [20]. The readmission rates for Hospital Compare are created by multiplying the national unadjusted rates by the predicted-to-expected ratio, with the predicted outcome (numerator) calculated using hospital-specific risk estimates and the expected outcome (denominator) calculated with

population-level risk estimates. PSI-90 is reported as a two-year average, and the 30-day hospital-wide readmission rate is reported as a three-year average, and, similar to previous literature, these data were assigned to the terminal year of the reporting period [21, 22]. These two quality metrics were chosen as proxies of overall hospital quality given the consistency of public reporting to Hospital Compare and the existing literature that focuses on quality performance related to readmission rates and patient safety [23–27]. In an effort to identify a threshold effect, the quality variables were divided into quintiles.

Additional covariates included rurality, teaching hospital status, number of hospital beds, system membership, ownership, Medicare payor mix, and market competition. Ownership was defined as non-profit, for-profit, or government. Medicare payor mix was defined as proportion of inpatient days from Medicare patients and was included in the model given the influence of Medicare reimbursement on hospital finances. The Hirschman-Herfindahl Index (HHI), calculated based on a hospital's share of annual discharges within its hospital referral region, was a proxy for the level of hospital market competition and incorporated to the model to adjust for geographic variation in reimbursement and expenses [28]. All dollar values were converted to 2018 U.S. dollars.

## Statistical analysis

We reported unadjusted hospital general and financial characteristics as medians and interquartile ranges and proportions where appropriate. For the regression analyses, we first performed multilevel mixed effects linear regressions with operating margin and total margin as the dependent variables and quality metrics and hospital characteristics as independent variables. Given concern for collinearity across the two quality variables, each model was run with independent variables that included the hospital characteristics and one of the two quality metrics (30-day hospital-wide readmission rate or PSI-90) from the same year as the dependent variable. Hospitals were grouped into quintiles of quality performance with the worst performers (highest readmission rates and highest PSI-90 scores) serving as the reference group. We created additional regression models with the same dependent variables but using the previous year's measured quality scores and hospital characteristics as independent variables (see **eMethods** in the S1 File for additional details).

Given the longitudinal design, we nested hospital-observations within hospitals (**S2 Table in S1 File**). To account for unobserved variables that are constant across hospitals but may vary over time, fixed year effects were included. For random effects, we used random intercepts at the hospital level. The longitudinal observations within each hospital are correlated, and random intercepts account for this clustering and provide an estimate of hospital-level variation. We did not include random slopes. To reduce potential confounding bias, covariate adjustment was incorporated into our models [29]. Goodness of fit of the mixed models was defined using the marginal $R^2$ and conditional $R^2$ as defined by Nakagawa and Schielzoth [30]. Second, we performed a mixed effects logistic regression with the same independent variables of the previous regressions and with financial distress as the dependent variable. This study did not involve personally identifiable information from human participants and did not require institutional review board approval. Given the multiple comparisons, we applied a Bonferroni correction and considered a two-tailed p value less than .017 to be statistically significant. All analyses were performed using Stata, version 15.1 (StataCorp, College Station, Texas).

## Results

Our sample included 20,919 observations from 4,331 unique hospitals (**S3 Table in S1 File**). In 2018, the median 30-day hospital wide readmission rate was 15.2 (IQR 14.8 to 15.6), the

**Table 1. General characteristics of hospitals in 2018.**

| No. Hospitals | | |
|---|---|---|
| **General Characteristics** | **No.** | **Summary** |
| Rural—no. (%) | 3357 | 1305 (38.9) |
| Teaching Hospital—no. (%) | 3357 | 1480 (44.1) |
| Hospital Bed Size—no. (%) | 3357 | |
| 1–99 | | 1673 (49.8) |
| 100–299 | | 1054 (31.4) |
| $\geq$ 300 | | 630 (18.8) |
| Member of Hospital System—no. (%) | 3357 | 2283 (68.0) |
| Ownership | 3357 | |
| Non-Profit | | 2204 (65.7) |
| For-Profit | | 468 (13.9) |
| Government (Non-Federal) | | 685 (20.4) |
| Medicare Payor Mix, median (IQR)[a] | 3357 | 54.7 (44.0–65.1) |
| Herfindahl Hirschman Index | 3357 | |
| Unconcentrated | | 1359 (40.5) |
| Moderate Concentration | | 761 (22.7) |
| High Concentration | | 1237 (36.8) |
| **Quality** | | |
| 30-Day Hospital Readmission Rate, median (IQR) | 3252 | 15.2 (14.8–15.6) |
| PSI-90 Safety Score, median (IQR) | 2389 | 0.96 (0.89–1.07) |
| **Finances** | | |
| Operating Margin, median (IQR) | 3137 | -1.8 (-9.7–5.9) |
| Total Margin, median (IQR) | 3126 | 4.3 (-0.8–10.2) |
| Working Capital to Total Assets, median (IQR) | 3306 | 0.22 (-0.02–0.58) |
| Retained Earnings to Total Assets, median (IQR) | 3306 | 0.67 (0.40–0.91) |
| EBIT to Total Assets, median (IQR) | 3306 | 0.11 (-0.01–0.59) |
| Total Equity to Total Liabilities, median (IQR) | 3306 | 0.93 (0.26–2.21) |
| Financial Distress—no. (%) | 3306 | 750 (22.7) |

Abbreviations: IQR = Interquartile range, EBIT = earnings before interest and taxes.

[a] Medicare Payor Mix represents Medicare days divided by total inpatient days.

median PSI-90 score was 0.96 (IQR 0.89 to 1.07), the median operating margin was -1.8 (IQR -9.7–5.9), and 750 (22.7%) hospitals experienced financial distress (Table 1). The median working capital to total assets (X1) was 0.22 (IQR -0.02 to 0.58), the median retained equity to total assets (X2) was 0.67 (IQR 0.40 to 0.91), the median earnings before interest and taxes to total assets (X3) was 0.11 (IQR -0.01 to 0.59), and the median total equity to total liabilities (X4) was 0.93 (IQR 0.26 to 2.21).

In the analyses with same-year quality data, top-performing quality hospitals experienced superior financial performance compared to the worst-performing hospitals. Hospitals within the lowest quintile of 30-day readmission rates (top-performing) experienced a higher operating margin (+0.95%, CI 0.51–1.39, $p < .001$) and higher total margin (+0.77%, CI 0.42–1.11, $p < .001$) compared to hospitals within the highest quintile (worst-performing) of 30-day readmission rates (Fig 1), full model results in **S4** and **S5 Tables in** S1 File). Hospitals within the lowest quintile of PSI-90 scores (top-performing) experienced a higher operating margin (+0.62%, CI 0.17–1.08, $p = .007$) and higher total margin (+0.52%, CI 0.16–0.89, $p = .004$) compared to hospitals within the highest quintile (worst-performing) of PSI-90 scores.

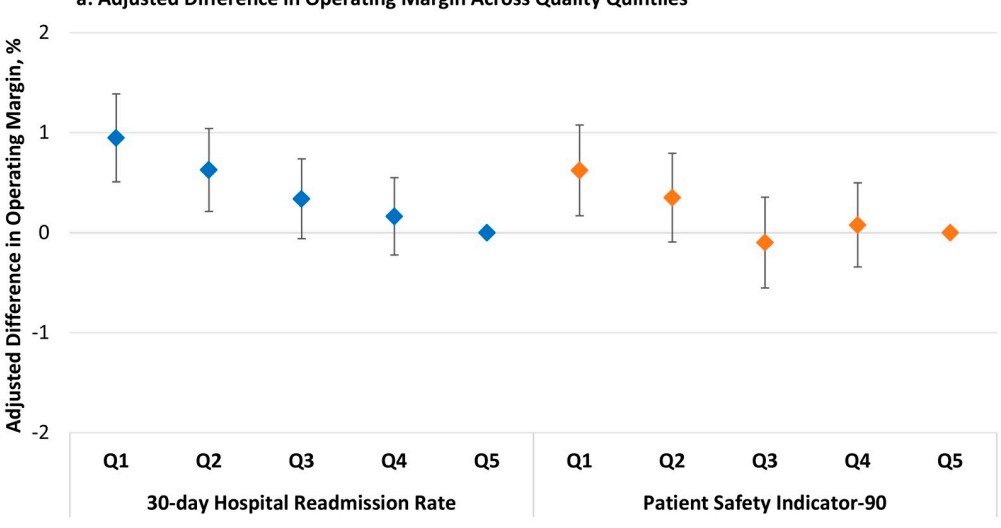

**Fig 1. Adjusted difference in operating and total margin across quality quintiles.** Hospitals within the lowest 30-day hospital-wide readmission rate quintile and hospitals within the lowest PSI-90 score quintile experienced higher (a) operating margins and (b) total margins compared to their respective highest quality quintiles. Abbreviations: Q = quintile Vertical bars are the 95% confidence intervals. Q5 (worst) is the reference quintile. 30-day Readmission Rates (median): Q1 = 14.6, Q2 = 15.1, Q3 = 15.3, Q4 = 15.6, Q5 = 16.3 Patient Safety Indicator-90 (median): Q1 = 0.75, Q2 = 0.85, Q3 = 0.91, Q4 = 0.96, Q5 = 1.08 For readmission rates and PSI-90 scores, higher values represent lower quality.

Similarly, in the analyses performed using the prior year's quality data, hospitals within the lowest quintile of 30-day readmission rates (top-performing) experienced a higher operating margin (+0.91%, CI 0.41–1.40], $p < .001$) and a higher total margin (+0.77%, CI 0.37–1.61, $p < .001$) in the subsequent year compared to hospitals within the highest quintile (worst-performing) of 30-day readmission rates (Fig 2), full model results in **S6** and **S7 Tables in** S1 File). Hospitals within the lowest quintile of PSI-90 scores (top-performing) experienced a higher operating margin (+0.66%, CI 0.15–1.16, $p = .011$) compared to hospitals within the highest quintile (worst-performing) of PSI-90 scores.

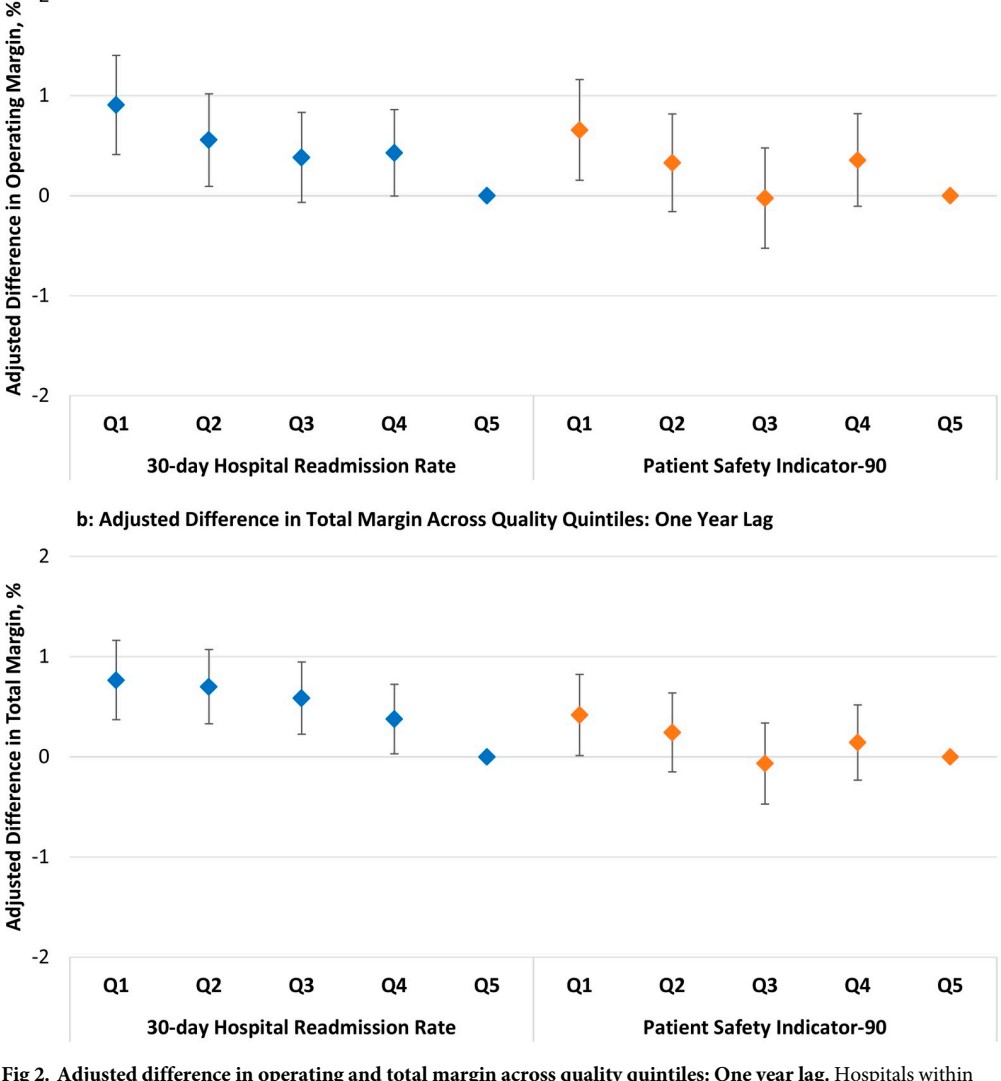

**Fig 2. Adjusted difference in operating and total margin across quality quintiles: One year lag.** Hospitals within the lowest previous year's 30-day hospital-wide readmission rate quintile and hospitals within the lowest previous year's PSI-90 score quintile experienced higher (a) operating margins and (b) total margins compared to their respective highest quality quintiles. Abbreviations: Q = quintile Vertical bars are the 95% confidence intervals. Q5 (worst) is the reference quintile. 30-day Readmission Rates (median): Q1 = 14.6, Q2 = 15.1, Q3 = 15.3, Q4 = 15.6, Q5 = 16.3 Patient Safety Indicator-90 (median): Q1 = 0.75, Q2 = 0.85, Q3 = 0.91, Q4 = 0.96, Q5 = 1.08 For readmission rates and PSI-90 scores, higher values represent lower quality.

In the analyses for financial distress, hospitals within the lowest quintile of 30-day readmission rates experienced lower odds of financial distress in the same year compared to hospitals within the highest quintile of 30-day readmission rates (Fig 3, adjusted odds ratio [aOR] 0.56, CI 0.45–0.70, p < .001), full model results in **S8 Table in** S1 File). Hospitals within the lowest quintile of PSI-90 experienced lower odds of financial distress in the same year compared to hospitals within the highest quintile (aOR 0.70, CI 0.56–0.89, p = .003). Simiarly, in the lag-year analysis for financial distress, hospitals within the lowest quintile of 30-day readmission rates experienced lower odds of financial distress in the subsequent year compared to hospitals within the highest quintile of 30-day readmission rates (Fig 4, aOR 0.65, CI 0.50–0.84,

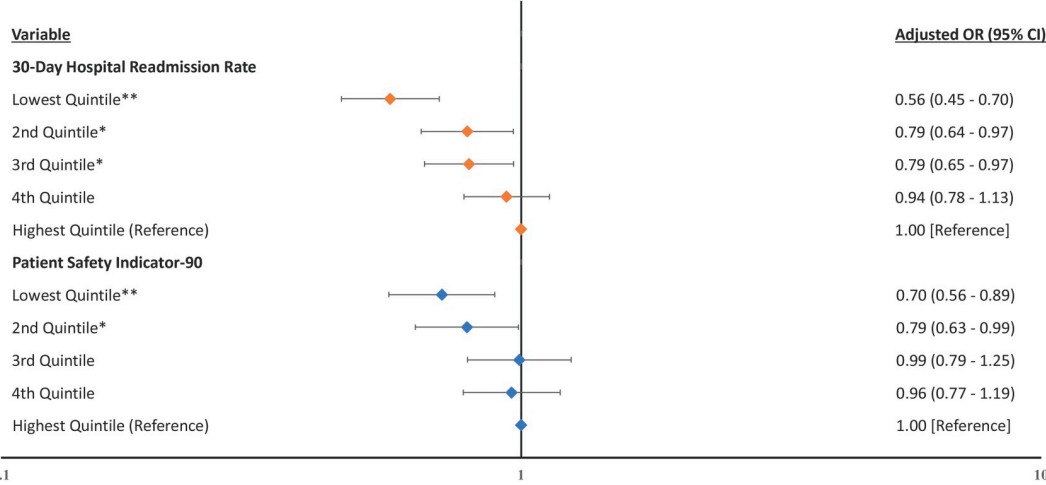

**Fig 3. Forest plot of the odds ratios for the association between same year's measured quality and financial distress.** Hospitals within the lowest 30-day hospital-wide readmission rate quintile and hospitals within the lowest PSI-90 score quintile experienced lower odds of financial distress compared to their respective highest quality quintiles. **Notes**: Forest plot showing the adjusted odds ratios with financial distress as the dependent variable with quality indicators as independent variables and adjusted for the hospital characteristics. Higher rates and scores represent worse quality. * p ≤ .05 ** p ≤ .01.

p = .001), full model results in **S9 Table in** S1 File). Hospitals within the lowest quintile of PSI-90 experienced lower odds of financial distress in the subsequent year compared to hospitals within the highest quintile (aOR 0.65, CI 0.49–0.85, p = .002).

## Discussion

Hospitals must balance their mandate to deliver excellent quality of care while also generating enough profits to keep their doors open. Unfortunately, the fee-for-service insurance model

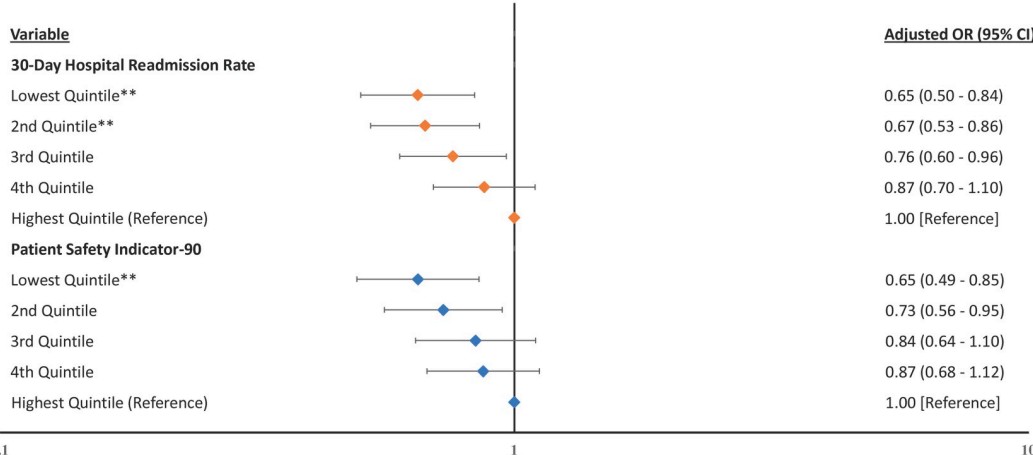

**Fig 4. Forest plot of the odds ratios for the association between previous year's measured quality and financial distress.** Hospitals within the lowest previous year's 30-day hospital-wide readmission rate quintile and hospitals within the lowest previous year's PSI-90 score quintile experienced lower odds of financial distress compared to their respective highest quality quintiles. **Notes**: Forest plot showing the adjusted odds ratios with financial distress as the dependent variable with quality indicators as independent variables and adjusted for the hospital characteristics. Higher rates and scores represent worse quality. ** p ≤ .01.

creates an environment in which hospitals may focus on the quantity of care and not pay sufficient attention to quality as the penalties associated with lower quality may not offset the incentives to increase lucrative procedural volume. To combat these misaligned incentives, CMS continues to refine and improve their public reporting and payment programs to align incentives between providers and patients [31]. Using hospital-level data from CMS' national publicly reported Hospital Compare program, we identified that higher measured quality was associated with superior financial performance and with lower odds of financial distress in the same year and the subsequent year. These findings suggest that a hospital's high-quality performance today may be associated with its future financial performance.

Our findings have important implications. First, our results build on previous work and provide further evidence in support of a link between high-quality care and superior hospital-level financial performance [32]. Among 108 acute care hospitals in New York state, Akinleye et al. (2019) identified a positive correlation between financial performance and hospital quality. Our study expands on their work and includes a national population of hospitals and multiple years of financial and quality data. Additionally, our study draws attention to the incentives in health care delivery. Revenue generation for many hospitals remains connected to fee-for-service reimbursement where hospitals have an incentive to drive high volumes of care to support their financial solvency. A 2018 study from the Health Care Payment Learning and Action Network, a subset of the Department of Health and Human Services, indicated that 41% of payments were volume-based and unrelated to quality performance [33]. While national level value-based programs have emerged over the past decade, there remains concern that the financial incentives for hospitals to profit through fee-for-service contribute to inefficient and poorly coordinated care [34].

Realigning our health system to produce high value for patients should be the defining goal of our health care delivery systems because focusing on value—defined as health outcomes achieved per dollar spent—has the potential to align incentives across the various stakeholders —patients, providers, insurers, and suppliers [35]. Initiatives like Hospital Compare and the Leapfrog Group's Annual Hospital Survey [36], which measure and report quality outcomes, can help move us more rapidly toward this goal. Previous evidence suggests that more than 10% of hospital readmissions are preventable [37], and increased transparency and communication regarding best practices may help us collectively deliver better value care to patients through reduced readmissions and fewer complications.

Second, our findings are consistent for both the same year and lag year analyses, and this suggests that excellent quality today may have a sustained and positive effect on a hospital's financial performance. We believe there at least two factors may underlie these findings. First, hospitals that provide excellent quality may have fewer complications and may have developed efficient systems to deliver care and avoid readmissions. High-quality hospitals may have fewer operating expenses related to managing the complications that constitute the PSI-90 score and may experience a lower probability of absorbing costly readmissions. Second, there may be a time dependent decrease in future revenue streams because the transparency around quality reporting may drive patients to avoid hospitals with inferior performance on measured quality metrics. A previous survey of 2,122 surgical patients suggested that when patients were choosing a future hospital for care delivery, those individuals who experienced adverse outcomes after discharge were more likely to pay attention to quality-of-care measures as compared to those that did not experience an adverse outcome [38].

Third, our findings suggest that targeting quality may be an avenue to avoid financial failure. Our study builds on previous literature that identified links between structural and operational characteristics and hospital financial distress [39–41]. In addition to the previously identified operational factors like occupancy and case mix, we recommend that hospital

managers consider quality as a modifiable factor to focus on as they look for opportunities to achieve positive financial performance and avoid financial distress. Outside of the possibility of bankruptcy, there are harmful consequences for organizations in financial distress, and hospitals are not an exception [42]. Weaker credit ratings, increased borrowing rates, and worse profit margins make it difficult for hospitals to recover once financial decline begins. Our findings offerevidence that hospitals can align the goals of delivering high-quality care and generating sustainable financial returns.

Our study includes important limitations. First, given the retrospective nature of the study, there is possible selection bias. Not every hospital reports data to the American Hospital Association and Hospital Compare databases, and our results may have limited generalizability to the hospitals that do not consistently report to these respective databases. For example, the 29% of observations missing PSI-90 may suggest that our findings have limited applicability to the hospitals that do not report PSI-90. While some may argue that imputation could help mitigate this limitation, we argue that imputing values for hospitals that lack PSI-90 scores involves overgeneralized assumptions that may not be accurate. Second, availability bias is an important concern. Our definition of hospital quality relies on what data have been publicly reported over the past decade. As described previously regarding quality composite indicators [43], this study utilizes the available data from Hospital Compare which may not be truly representative of hospital-wide quality but rather reflects what metrics are available for analysis. The underlying condition-specific Hospital Compare measures are adjusted based on demographic and clinical characteristics [20], but the adjustments may fail to account for additional clinical factors and non-clinical factors (e.g. socioeconomic status) that may be related to a measured outcome (e.g. readmission rate). Third, the quality variables are reported as a multi-year average, and our assumption of using the terminal year to represent the value in that year underestimates the degree of variance within those variables. Fourth, we acknowledge that an alternative conclusion is that hospitals with more available financing are better able to deliver high-quality care. Our study design does not allow for us to draw causal claims. We acknowledge that an association between measured quality and financial performance does not provide definitive evidence that quality improvement will lead to financial gain. However, our findings offer a foundation for future investigation into the association between quality improvement and financial performance. Fifth, there is a known limitation of omitted variable bias for models involving random effects. We incorporated fixed time effects to reduce the bias in our estimates related to unobserved time-invariant characteristics. Our models incorporated various hospital, geographic, and local market characteristics, which suggests some internal validity of our findings.

In conclusion, our study highlights the value of delivering excellent quality care. Hospitals that delivered high-quality care experienced superior financial returns in the same year and subsequent year. This virtuous cycle contrasts with the association of poor quality and financial distress, signaling to hospitals that quality may be important for the bottom line. To deliver better patient care, we should increase investments in reporting systems, refine the granularity of quality metrics, and enhance the effectiveness of hospital quality programs.

## Supporting information

**S1 File.**
(DOCX)

## Author Contributions

**Conceptualization:** Samuel J. Enumah, David C. Chang.

**Data curation:** Samuel J. Enumah.

**Formal analysis:** Samuel J. Enumah, David C. Chang.

**Funding acquisition:** Samuel J. Enumah, David C. Chang.

**Investigation:** Samuel J. Enumah, David C. Chang.

**Methodology:** Samuel J. Enumah, David C. Chang.

**Project administration:** Samuel J. Enumah.

**Resources:** Samuel J. Enumah, David C. Chang.

**Software:** Samuel J. Enumah.

**Supervision:** Andrew S. Resnick, David C. Chang.

**Visualization:** Samuel J. Enumah.

**Writing – original draft:** Samuel J. Enumah, David C. Chang.

**Writing – review & editing:** Samuel J. Enumah, Andrew S. Resnick, David C. Chang.

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
