## [Decision Letter · Decision Letter 0]

15 Mar 2022

PONE-D-22-05106Association of measured quality with financial health among U.S. hospitalsPLOS ONE

Dear Dr. Enumah,

Thank you for submitting your manuscript to PLOS ONE. After careful consideration, we feel that it has merit but does not fully meet PLOS ONE’s publication criteria as it currently stands. Therefore, we invite you to submit a revised version of the manuscript that addresses the points raised during the review process.

We look forward to receiving your revised manuscript.

Kind regards,

María del Carmen Valls Martínez, Ph.D.

Academic Editor

PLOS ONE

Journal Requirements:

Reviewers' comments:

Reviewer's Responses to Questions

**Comments to the Author**

1. Is the manuscript technically sound, and do the data support the conclusions?

Reviewer #1: Partly

Reviewer #2: Yes

Reviewer #3: Yes

2. Has the statistical analysis been performed appropriately and rigorously? 

Reviewer #1: Yes

Reviewer #2: Yes

Reviewer #3: Yes

3. Have the authors made all data underlying the findings in their manuscript fully available?

Reviewer #1: Yes

Reviewer #2: No

Reviewer #3: Yes

4. Is the manuscript presented in an intelligible fashion and written in standard English?

Reviewer #1: Yes

Reviewer #2: Yes

Reviewer #3: Yes

5. Review Comments to the Author

Reviewer #1: 1- In the whole text of the article, the sentences are mostly long, and it is suggested to write more academically to understand the concept better.

2- The method part of the article is very detailed

3-The charts and tables' quality is low, and they are not legible. It is better to be modified in the original version

Reviewer #2: Dear authors, i would like to thank you for your such amazing work to measure the association of quality with financial health among U.S hospitals. However, minor comments that needs clarification before accepting this paper.

The paper is well written, concise, and logic.

Abstract:

-Observational study or retrospective study design?

-Please define your abbreviation such as IQR... etc.

- The first paragraph in result section should be in method section.

Introduction

- Please remove the word of relationship since you use the word of association in all text.

- Please at the last of introduction, summarize all of your study aims.

Material and methods

- Please report that all 4331 hospitals have enter their data, What about missing numbers? why you excluded both psychiatric and pediatric hospitals? add your study design.

- 30-readmission rate? Is it standard?

- Analysis part is perfect.

Discussion

-The first paragraph are perfect and i recommend to add your findings on it.

- Lack of comparing your result with previous results, more literature review will enhance your paper.

- What should managers of hospital do? If you add recommendation for mangers would be great.

References

-Please return to PLOS One guidelines if they reported DOI with references.

- Double check on your references. For example reference # 10 page number is missing.

Figures

All figures and footnote are unclear, blurry, and hard to read.

Reviewer #3: The topic under study is quite a critical one and the results revealed are a bonus. A generally well written paper that meets the necessary requirements of scientific writing. In terms of presentation of results, I personally find the figures a bit difficult to read, which could be improved.

6. PLOS authors have the option to publish the peer review history of their article (what does this mean?). If published, this will include your full peer review and any attached files.

Reviewer #1: No

Reviewer #2: No

Reviewer #3: No

---

## [Author Response · Author response to Decision Letter 0]

22 Mar 2022

David C. Chang, PhD, MPH, MBA

Associate Professor of Surgery 

Massachusetts General Hospital

Codman Center for Clinical Effectiveness in Surgery

165 Cambridge Street, Suite 403

Boston, MA 02114

Dear Dr. del Carmen Valls Martínez, 

Thank you for the opportunity to submit a revised version of our manuscript entitled “Association of measured quality with financial health among U.S. hospitals” for consideration as an original research article in PLOS ONE.

We appreciate the comments and critiques provided by the Reviewers to enhance our manuscript’s quality. We have pasted their comments below and provided responses to each of their comments in red. As requested, we have attached both a clean version and tracked changes version of our manuscript for further review. Please do not hesitate to contact us with any questions or concerns.

Sincerely,

David C. Chang, PhD, MPH, MBA

Associate Professor of Surgery, Massachusetts General Hospital

Codman Center for Clinical Effectiveness in Surgery

Email: dchang8@mgh.harvard.edu / Phone: 617-643-6730

 

Reviewers’ Comments to the Author

Reviewer #1: 1- In the whole text of the article, the sentences are mostly long, and it is suggested to write more academically to understand the concept better.

Thank you for this comment. We have revised the manuscript to create more concise sentences and to add clarity.

2- The method part of the article is very detailed

Thank you for this comment. Our aim in this section was to provide the reader with a thorough understanding of our approach and process in generating our findings. 

3-The charts and tables' quality is low, and they are not legible. It is better to be modified in the original version

Thank you for this comment. Our sincere apologies for the issue with legibility. We have included higher quality TIFF images with our re-submission.

Reviewer #2: Dear authors, i would like to thank you for your such amazing work to measure the association of quality with financial health among U.S hospitals. However, minor comments that needs clarification before accepting this paper.

The paper is well written, concise, and logic.

Abstract:

-Observational study or retrospective study design?

Retrospective observational cohort study. We have changed the abstract. Thank you for this comment.

-Please define your abbreviation such as IQR... etc.

IQR = Interquartile range. We have changed the abstract. Thank you for this comment.

- The first paragraph in result section should be in method section.

We moved the first sentence of this part of the Abstract Results to the Abstract’s Methods section. 

Thank you for this comment.

Introduction

- Please remove the word of relationship since you use the word of association in all text.

Thank you for this comment. We have replaced “relationship” with association.

- Please at the last of introduction, summarize all of your study aims.

Thank you for this comment. We have added an aim to our first sentence of the final paragraph in our Introduction.

Material and methods

- Please report that all 4331 hospitals have enter their data, What about missing numbers? why you excluded both psychiatric and pediatric hospitals? add your study design.

Thank you for these comments. Our S1 Table provides missingness information for the included variables. Our S3 Table in the appendix provides information about hospitals and the observation-years contributed by hospitals. Some hospitals contributed in every year while others contributed fewer observation-years. We excluded psychiatric and pediatric hospitals and long-term care facilities as these facilities have different reimbursement structures compared to acute care hospitals and may not have similarly defined quality goals.

-We have added clarification to the Introduction regarding why certain types of facilities were excluded.

-We have added clarification to the Introduction on the study design.

- 30-readmission rate? Is it standard?

Thank you for this comment. Yes, this is a standard measure of quality. It is risk-standardized by the Centers for Medicare & Medicaid Services (CMS). It is derived from “a composite of seven statistical models that are built for groups of admissions that are clinically related.”

For further details, please see this online reference from CMS:

https://www.cms.gov/Medicare/Quality-Initiatives-Patient-Assessment-Instruments/MMS/Downloads/MMSHospital-WideAll-ConditionReadmissionRate.pdf

We have added “risk-standardized” language to the Methods to clarify this.

- Analysis part is perfect.

Thank you for this comment.

Discussion

-The first paragraph are perfect and i recommend to add your findings on it.

Thank you for this comment. We have aimed in our revisions to clarify our findings.

- Lack of comparing your result with previous results, more literature review will enhance your paper.

Thank you for this comment. We have added more detail about the previous literature that does exist. We added:

“Among 108 acute care hospitals in New York state, Akinleye et al. (2019) identified a positive correlation between financial performance and hospital quality. Our study expands on their work and includes a national population of hospitals and multiple years of financial and quality data.”

Overall, there is little available data on this topic, and we believe this helps stress the importance of our disseminating our findings.

- What should managers of hospital do? If you add recommendation for mangers would be great.

Thank you for this comment. We have added a concise and clear recommendation for managers.

“In addition to the previously identified operational factors like occupancy and case mix, we recommend that hospital managers may consider quality as a modifiable factor to focus on as they look for opportunities to achieve positive financial performance and avoid financial distress.”

References

-Please return to PLOS One guidelines if they reported DOI with references.

Thank you for this comment. We have added DOI where it was indicated.

- Double check on your references. For example reference # 10 page number is missing.

Thank you for this comment. We have reviewed the references and made the appropriate changes.

Figures

All figures and footnote are unclear, blurry, and hard to read.

Thank you for this comment. We apologize for the issue with legibility.

We have included higher quality TIFF images with our re-submission.

Reviewer #3: The topic under study is quite a critical one and the results revealed are a bonus. A generally well written paper that meets the necessary requirements of scientific writing. In terms of presentation of results, I personally find the figures a bit difficult to read, which could be improved.

Thank you for this comment. We apologize for the issue with legibility.

We have included higher quality TIFF images with our re-submission.

---

## [Editor Report · Decision Letter 1]

25 Mar 2022

Association of measured quality with financial health among U.S. hospitals

PONE-D-22-05106R1

Dear Dr. Samuel Joseph Enumah,

We’re pleased to inform you that your manuscript has been judged scientifically suitable for publication and will be formally accepted for publication once it meets all outstanding technical requirements.

Kind regards,

María del Carmen Valls Martínez, Ph.D.

Academic Editor

PLOS ONE

---

## [Editor Report · Acceptance letter]

30 Mar 2022

PONE-D-22-05106R1 

Association of measured quality with financial health among U.S. hospitals 

Dear Dr. Enumah:

I'm pleased to inform you that your manuscript has been deemed suitable for publication in PLOS ONE. Congratulations! Your manuscript is now with our production department. 

Kind regards, 

on behalf of

Dr. María del Carmen Valls Martínez 

Academic Editor

PLOS ONE